# Effect of Angiotensin II on Chondrocyte Degeneration and Protection via Differential Usage of Angiotensin II Receptors

**DOI:** 10.3390/ijms22179204

**Published:** 2021-08-25

**Authors:** Takashi Nishida, Sho Akashi, Masaharu Takigawa, Satoshi Kubota

**Affiliations:** 1Department of Biochemistry and Molecular Dentistry, Okayama University Graduate School of Medicine, Dentistry and Pharmaceutical Sciences, 2-5-1 Shikata-cho, Kita-ku, Okayama 700-8525, Japan; de20002@s.okayama-u.ac.jp (S.A.); kubota1@md.okayama-u.ac.jp (S.K.); 2Advanced Research Center for Oral and Craniofacial Sciences, Okayama University Graduate School of Medicine, Dentistry and Pharmaceutical Sciences, 2-5-1 Shikata-cho, Kita-ku, Okayama 700-8525, Japan; takigawa@md.okayama-u.ac.jp

**Keywords:** angiotensin II, cellular communication network factor 2 (CCN2), renin–angiotensin system (RAS), losartan, angiotensin II type I receptor (AT_1_R)

## Abstract

The renin–angiotensin system (RAS) controls not only systemic functions, such as blood pressure, but also local tissue-specific events. Previous studies have shown that angiotensin II receptor type 1 (AT_1_R) and type 2 (AT_2_R), two RAS components, are expressed in chondrocytes. However, the angiotensin II (ANG II) effects exerted through these receptors on chondrocyte metabolism are not fully understood. In this study, we investigated the effects of ANG II and AT_1_R blockade on chondrocyte proliferation and differentiation. Firstly, we observed that ANG II significantly suppressed cell proliferation and glycosaminoglycan content in rat chondrocytic RCS cells. Additionally, ANG II decreased CCN2, which is an anabolic factor for chondrocytes, via increased MMP9. In *Agtr1a*-deficient RCS cells generated by the CRISPR-Cas9 system, *Ccn2* and *Aggrecan* (*Acan*) expression increased. Losartan, an AT_1_R antagonist, blocked the ANG II-induced decrease in CCN2 production and *Acan* expression in RCS cells. These findings suggest that AT_1_R blockade reduces ANG II-induced chondrocyte degeneration. Interestingly, AT_1_R-positive cells, which were localized on the surface of the articular cartilage of 7-month-old mice expanded throughout the articular cartilage with aging. These findings suggest that ANG II regulates age-related cartilage degeneration through the ANG II–AT_1_R axis.

## 1. Introduction

Obesity is a risk factor for osteoarthritis (OA) [1]. Overloading the knee joints due to obesity is considered a significant risk factor for the induction of articular cartilage degradation [1]. In addition to body fat, the infrapatellar fat pad, which is an adipose tissue in the knee joint, is increased with aging and obesity [2], and it is thought that this tissue secretes various inflammatory cytokines and adipokines, which may be involved in chondrocyte proliferation, differentiation, or degradation [2]. Therefore, the current idea that obesity only contributes to OA by overloading the knee joints is insufficient to account for the true pathophysiology of aging- and obesity-induced OA. Because it was reported that angiotensin II (ANG II) is one of the peptide hormones induced by aging and obesity [3] and that chondrocytes express it two specific receptors, angiotensin II receptor type 1 (AT_1_R) and type 2 (AT_2_R) [4], we focused on ANG II functions in chondrocytes in this study. In general, ANG II is synthesized via two cleavage steps [5]. Specifically, angiotensin I (ANG I), which is the precursor of ANG II, is formed by the cleavage of angiotensinogen produced in the liver by renin expressed in the kidney, and then ANG I is cleaved by angiotensin-converting enzyme 1 (ACE1) to form ANG II [5]. This classical system is called the systemic renin–angiotensin system (RAS) and encompasses the multiple organs involved in forming ANG II [5]. ANG II mainly acts through AT_1_R and AT_2_R [6]. AT_1_R is responsible for most of the biological effects of ANG II, and ANG II acts not only on blood vessels to stimulate constriction but also on the adrenal glands to stimulate the release of aldosterone [5,6]. As a result, ANG II substantially increases blood pressure through both increased vasoconstriction and increased reabsorption of salt and water [5,6]. Thus, the blockade of the systemic RAS has become an important clinical strategy in the treatment of renal and cardiovascular diseases, such as hypertension, heart failure, and diabetic nephropathy [5,6]. On the other hand, the production of angiotensinogens was reported to occur not only in the liver but also in adipose tissues [7]. Additionally, it has been demonstrated that prorenin, which is an enzymatically inactive precursor of renin, is present in various organs, including heart, kidney, and fat tissues, and circulates in the blood [8]. Prorenin binds to the prorenin receptor (PRR) [9,10,11,12], which induces a conformational change in prorenin [13], leading to the activation of prorenin. Then, activated prorenin cleaves angiotensinogen to form ANG I in local tissues, thus suggesting that “local” tissue-specific ANG II is produced and exerts various biological effects in different tissues [14]. In fact, it was reported that RAS components, including prorenin, PRR, and ACE1, were expressed in articular chondrocytes [15] and that cartilage-derived ANG II modulated the hypertrophic differentiation of chondrocytes [16]. Moreover, a recent study demonstrated that AT_1_R blockade reduced the chronic kidney disease-induced deterioration of bone [17]. These findings suggest that ANG II produced by the local RAS may be involved in the regulation of cartilage and bone metabolism.

Cellular communication network factor 2 (CCN2) is a matricellular protein with a molecular weight of 36–38 kDa and is involved in multiple biological events, such as development, wound healing, and tissue regeneration [18,19,20]. Our previous studies using an in vitro culture system demonstrated that CCN2 promoted the proliferation and differentiation of chondrocytes [21] and osteoblasts [22] and the formation of osteoclasts [23]. Furthermore, our in vivo study revealed that intraarticular injection of CCN2 enhanced the repair of articular cartilage in mono-iodoacetate (MIA)-induced OA-like lesions [24]. These findings suggest that CCN2 has anabolic functions in skeletal tissues, such as cartilage and bone. Notably, in muscle and renal cells, some studies have demonstrated that ANG II mediates the induction of CCN2 production and is associated with fibrosis [25,26]. However, the effect of ANG II on chondrocytes, particularly through the production of CCN2, is still unknown. Therefore, in this study, we analyzed the effects of ANG II and AT_1_R blockade on the proliferation, differentiation, and functional disorder of chondrocytes and explored the involvement of ANG II in the production of CCN2 in chondrocytes.

## 2. Results

### 2.1. Gene Expression of Local RAS Components in HCS-2/8 Cells Treated with MIA

As our previous study showed that the degradation of the cartilage matrix is induced by intraarticular injection of MIA, which inhibits glycolysis [24], we first investigated the gene expression levels of local RAS components in a human chondrosarcoma-derived chondrocytic cell line, HCS-2/8 [27], treated with MIA at concentrations of 2 and 4 μg/mL. As shown in Figure 1A, the gene expression levels of *ACE1*, *AGTR1,* and *AGTR2,* but not *ATP6AP2,* which encodes PRR, were significantly increased by treatment with 4 μg/mL MIA. These results indicate that RAS components expressed in chondrocytes are upregulated by cartilage degeneration, thus suggesting that the production of ANG II is promoted during the degeneration of cartilage.

### 2.2. Gene Expression of Local RAS Components by ANG II

Next, we examined whether ANG II regulates the gene expression of local RAS components. When the gene expression levels of local RAS components were examined in HCS-2/8 cells stimulated with 200 nM ANG II, both *AGTR1* and *AGTR2* expression levels were significantly increased (Figure 1B). These findings suggest that after ANG II is increased by cartilage degradation, it may amplify its signaling in chondrocytes in an autocrine/paracrine manner.

### 2.3. Effect of ANG II on Chondrocyte Proliferation and Differentiation

Next, we investigated the effect of ANG II on chondrocyte proliferation and differentiation. As shown in Figure 1C, ANG II at a concentration of 250 nM decreased the cell proliferation of a rat chondrosarcoma (RCS) cell line [28]. Furthermore, at the same concentration, ANG II also decreased the accumulation of glycosaminoglycans (GAGs) in RCS cells (Figure 1D). These results indicate that ANG II inhibits the proliferation and differentiation of chondrocytes, suggesting that ANG II negatively regulates cartilage metabolism.

### 2.4. Production of CCN2 and MMP9 in the Presence of ANG II

It has been reported that ANG II increases CCN2 production in renal and skeletal muscle cells [25,26]. Therefore, to confirm the ANG II-induced production of CCN2 in chondrocytes, we performed Western blot analysis. As shown in Figure 2A,B, the production of CCN2 protein in HCS-2/8 cells treated with ANG II at concentrations of 50, 100, 150, 200, and 250 nM was decreased in a dose-dependent manner by ANG II concentrations up to 100 nM, and, subsequently, it tended to be increased with the two higher concentrations. Based on these results, we hypothesized that the ANG II-induced change in the CCN2 level was due to post-translational regulation, and we next analyzed the protein production of matrix metalloproteinase 9 (MMP9), which is one of the enzymes that cleave CCN2. Production of MMP9 was increased by ANG II concentrations up to 150 nM, whereas it decreased at higher concentrations (Figure 2B). These results indicate a reverse correlation between CCN2 and MMP9 production, suggesting that the CCN2 level is potentially modulated by ANG II via MMP9.

### 2.5. Signaling Pathway of ANG II in RCS Cells

Next, we confirmed whether intracellular signaling was activated by treatment with ANG II in RCS cells. As shown in Figure 2C, phosphorylation of extracellular signal-regulated kinase (ERK)1/2 was increased by treatment with ANG II at concentrations of both 100 and 200 nM. This result suggests that stimulation with ANG II activates intracellular signaling of chondrocytes through AT_1_R and AT_2_R.

### 2.6. Generation of Agtr1a-Deficient RCS Cells and the Effect on Chondrocyte Proliferation and Differentiation

It has been previously established that ANG II has two types of receptors, namely, AT_1_R and AT_2_R [5,6,7], and that AT_1_R predominates in chondrocytes [15]. Accordingly, we generated *Agtr1a*-deficient RCS cells using the CRISPR-Cas9 system and investigated the effect of ANG II on the proliferation and differentiation of these RCS cells. Using genomic PCR and DNA sequencing, we confirmed the deletion of bases 489–672 in exon 3 of *Agtr1a* mRNA (Figure 3A). To exclude an off-target effect, we established two clonal cells and showed that expression of *Atgr1a* was not detected at the mRNA or protein level in either cell type (Figure 3B,C). These data indicate that the *Atgr1a* gene was knocked out in the two clonal cells established by the CRISPR-Cas9 system (KO1 and KO2 cells). Using these KO1 and KO2 cells, we investigated the effect of *Agtr1a* deficiency on the proliferation of RCS cells. As shown in Figure 3D, cell proliferation of KO1 and KO2 cells was suppressed compared with that of wild-type (WT) cells. Additionally, when KO1 and KO2 cells reached confluence, the metachromatic properties of toluidine blue staining were enhanced in both KO1 and KO2 cell cultures compared with that in WT cell cultures (Figure 3E). Moreover, the gene expression levels of two major markers of chondrocytes, type II collagen (*Col2a1*) and aggrecan (*Acan*), and chondrocyte anabolic factor, *Ccn2*, were examined. As a result, *Col2a1* expression was decreased, and conversely, *Ccn2* expression was increased in KO1 and KO2 cells compared with those in WT cells (Figure 3F). The *Acan* expression level was higher in KO2 cells than that in WT cells, but it did not differ between KO1 and WT cells (Figure 3F). Furthermore, when KO1 and KO2 cells were treated with ANG II, *Col2a1* expression was upregulated, but *Acan* expression was not affected (Figure 3G).

### 2.7. Effect of AT_1_R or AT_2_R Blockade on Col2a1, Acan, and Ccn2 Expression

Based on the results in Figure 3, we hypothesized that ANG II increases *Col2a1* expression via AT_2_R and decreases *Acan* expression via AT_1_R. To verify this hypothesis, we investigated *Col2a1* and *Acan* expression levels in RCS cells treated with losartan, an AT_1_R antagonist, and PD123319, an AT_2_R antagonist. The gene expression levels of *Col2a1* and *Acan* were significantly upregulated and downregulated by treatment with ANG II, respectively (Figure 4A). On the other hand, *Col2a1* expression was significantly increased by treatment with losartan, but *Acan* expression remained unchanged when RCS cells were stimulated with ANG II (Figure 4B). In contrast, PD123319 treatment significantly decreased *Acan* expression, and *Col2a1* expression remained unchanged (Figure 4C). Taken together with the data in Figure 3, considering that both AT_1_R and AT_2_R are expressed in chondrocytes, these results suggest that ANG II increases *Col2a1* expression via AT_2_R and decreases *Acan* expression via AT_1_R. On the other hand, *Ccn2* expression was not changed by ANG II in RCS cells treated with either losartan or PD123319 (Figure 4). These results indicate that CCN2 production is principally modulated by ANG II via MMP9, not directly by ANG II.

### 2.8. Effect of AT_1_R Blockade on CCN2 Production

Based on the results of Figure 2, Figure 3 and Figure 4, we hypothesized that the ANG II–AT_1_R pathway suppresses the production of CCN2 via MMP9. To test this hypothesis, we investigated whether or not the production of CCN2, which is one of the anabolic factors for chondrocytes, was altered by the addition of losartan. As shown in Figure 5A,B, ANG II decreased CCN2 production in RCS cells in the absence of losartan, but not in its presence. Furthermore, we confirmed that ANG II-induced MMP9 had gelatinase activity in the culture medium (Figure 5C). The bands of active MMP9 were increased by ANG II at 100 nM and 250 nM, which is consistent with the results in Figure 2B. Losartan decreased ANG II-induced gelatinase activity (Figure 5C). On the other hand, the bands of MMP2 were not affected by ANG II (Figure 5C). This result suggests that AT_1_R blockade may stabilize CCN2 by decreasing MMP9, thus supporting the idea that AT_1_R blockade may be useful for the suppression of ANG II-induced cartilage degeneration.

### 2.9. Regulation of Gene Expression of Local RAS Components by Age

We next investigated whether or not the gene expression levels of local RAS components change with age. We obtained the xiphoid process from mice at 7 and 13 months of age and subsequently isolated chondrocytes from its tissue [29]. We found that *Ccn2* expression therein was decreased with age (Figure 6A). The gene expression level of *Atp6ap2*, which encodes PRR, tended to be upregulated with age (*p* = 0.051) (Figure 6B). *Ace1* and *Agtr1* expression showed no change with age (Figure 6C,D), but *Agtr2* expression was significantly increased in older mice (Figure 6E).

### 2.10. Localization of AT_1_R in Articular Cartilage According to Age

We next used immunohistochemistry to comparatively analyze the localization of AT_1_R in the articular cartilage of tibias of 7- and 13-month-old mice. As shown in Figure 7, toluidine blue staining showed metachromasia in the articular cartilage of 7-month-old mice, and AT_1_R was detected in the superficial zone of articular cartilage. In aged mice (13-month-old mice), toluidine blue staining was remarkably decreased in articular cartilage. This finding suggests that the proteoglycan of articular cartilage is lost during aging. However, AT_1_R-positive cells expanded throughout the articular cartilage in aged mice (Figure 7). These results suggest that AT_1_R-positive cells expand to the deeper zones of articular cartilage with age, although the *Agtr1* expression in each cell is unchanged (Figure 6D).

## 3. Discussion

In this study, we observed that ANG II decreased chondrocyte proliferation, *Acan* expression, and the accumulation of GAG content (Figure 1). These results suggest that ANG II suppresses the proliferation and differentiation of chondrocytes. Regarding chondrocyte hypertrophy, a previous study using ATDC5 cells reported that ANG II inhibited the gene expression of *Col10a1*, which is a marker of hypertrophic chondrocytes [16]. In the same report, the authors demonstrated that *ColX10a1* expression was upregulated and downregulated by treatment with olmesartan, which is an inhibitor of AT_1_R, and PD123319, which is an inhibitor of AT_2_R, respectively [16]. Based on these findings, the authors suggested that the activation of AT_1_R by ANG II suppresses differentiation into hypertrophic chondrocytes, whereas activation of AT_2_R by ANG II enhances it [16]. Notably, it was reported that *AGTR1* expression was detected not only in articular chondrocytes derived from OA and rheumatoid arthritis (RA) patients but also in normal chondrocytes, whereas *AGTR2* expression was detected in diseased chondrocytes only [15]. These results indicate that AT_1_R is the dominant ANG II receptor in chondrocytes, and AT_2_R is a transient receptor expressed under degradative conditions, suggesting that AT_1_R and AT_2_R not only modulate differentiation into hypertrophic chondrocytes but also contribute to chondrocyte degradation in OA and RA. Indeed, our data indicate that *Agtr1* expression did not differ between chondrocytes from mice at 7 and 13 months of age, whereas *Agtr2* expression was significantly upregulated in those from the aged mice (Figure 6). Additionally, we observed that MMP9 production was increased by ANG II at low concentrations in cultured chondrocytes (Figure 2). These results suggest that a low dose of ANG II during the early stage of cartilage degradation increases MMP9 production via AT_1_R. Considering that MMP9 is an important enzyme causing cartilage degradation and that AT_1_R is the dominant ANG II receptor in chondrocytes, ANG II-AT_1_R signaling may be involved in the degeneration of chondrocytes. Indeed, when HCS-2/8 cells were degenerated by MIA treatment, the gene expression levels of *ACE1*, which is a key component of the RAS, and those of both *AGTR1* and *AGTR2* were significantly upregulated (Figure 1A). These results indicate that the production of ANG II is increased in degenerated chondrocytes, thus suggesting that ANG II potentially accelerates chondrocyte degeneration induced by MIA. Moreover, we observed that CCN2 was not decreased in chondrocytes pre-treated with losartan (Figure 5). As we previously demonstrated that the injection of CCN2 into the knee joint promoted the repair of articular cartilage degenerated by MIA [24], we suggest that CCN2 has a therapeutic effect on chondrocyte degeneration. Therefore, the result showing that CCN2 production was maintained by losartan treatment indicates that AT_1_R blockade plays an important role in the suppression of chondrocyte degeneration. To confirm this hypothesis, we generated *Atgr1a*-deficient RCS cells by using the CRISPR-Cas9 system and investigated the effect of *Atgr1a* deficiency (Figure 3). When *Atgr1a*-deficient RCS cells were stimulated by ANG II, *Col2a1* expression was significantly increased similarly to that in WT chondrocytes, and *Acan* expression was not decreased compared to that of WT chondrocytes (Figure 3). As ANG II could activate AT_2_R in *Atgr1a*-deficient RCS cells, the action of ANG II via AT_2_R may have a suppressive effect on the loss of proteoglycan, leading to cartilage degeneration. These findings suggest that ANG II–AT_2_R signaling is potentially involved in the inhibition of chondrocyte degeneration. The results of using antagonists of AT_1_R and AT_2_R support these findings (Figure 4). Taken together, our results suggest that losartan, an anti-hypertensive drug, may reduce the risk of articular cartilage degradation.

It is well-known that ANG II regulates ECM accumulation in conditions such as fibrosis in the kidney and skeletal muscle, and CCN2 is considered to be one of the candidates for the effector molecules of fibrosis [25,26]. Several studies have reported that the ANG II–CCN2 axis plays an important role via AT_1_R in the development of renal and muscle fibrosis, along with increased type I collagen production [25,26]. On the other hand, our data reveal that in chondrocytes, ANG II decreased *Acan* expression and the accumulation of GAG content, but it increased *Col2a1* expression (Figure 1). Because the cartilage matrix consists of fibrous components (such as types II, IX, and XI collagens) to provide tensile strength and includes gel components (such as proteoglycan) to accommodate water [30], the suppressive effect of ANG II on *Acan* expression leads to a disorganized cartilage matrix. Therefore, we suggest that this suppressive effect of ANG II on cartilage matrix formation does not contradict the promotive effect of ANG II on fibrotic remodeling with type I collagen in other tissues. Furthermore, we observed that ANG II treatment did not affect *Ccn2* expression in WT RCS cells, although *Ccn2* expression was increased in *Atgr1a*-deficient cells (Figure 3 and Figure 4). These findings indicate that ANG II does not substantially regulate CCN2 expression in chondrocytes, contradicting previous studies that showed increased CCN2 expression through AT_1_R in renal and muscle cells [25,26]. We suspected that *Ccn2* expression is downregulated by destructive effects on the cartilage matrix via ANG II–AT_1_R signaling and is upregulated by protective effects via ANG II-AT_2_R signaling. However, *Ccn2* expression remained unchanged when RCS cells were treated with either losartan or PD123319 in the presence of ANG II (Figure 4). As we identified a reverse correlation between ANG II-regulated CCN2 and MMP9 production (Figure 2), we propose that CCN2 production is modulated via increased or decreased MMP9 production by ANG II. However, further investigation is needed to clarify the mechanism.

To investigate the degenerative change in articular cartilage accompanied by aging, we compared the articular cartilage of the tibias of 7-month-old mice with those of 13- month-old mice. Toluidine blue staining was markedly decreased in the articular cartilage of the tibias of mice at 13 months of age (Figure 7). On the other hand, compared with the localization at 7 months of age, AT_1_R was distributed throughout the articular cartilage of 13-month-old mice (Figure 7). These findings indicate that the degenerative change in articular cartilage with aging causes the expansion of localized AT_1_R, suggesting that aging amplifies ANG II-induced degeneration of articular cartilage via AT_1_R.

In conclusion, ANG II produced by the local RAS among chondrocytes suppresses the proliferation and differentiation of chondrocytes through AT_1_R and thus leads to chondrocyte degeneration (Figure 8A). As chondrocyte degeneration progresses, AT_2_R expression is increased in chondrocytes to protect the cell from degradation. ANG II promotes the gene expression of *Col2a1,* and ANG II at a high concentration increases the CCN2 level by decreasing MMP9 via AT_2_R and, as a result, exhibits cartilage protective effects, leading to the suppression of chondrocyte degeneration (Figure 8B). Therefore, AT_1_R blockers, such as anti-hypertensive drugs, may contribute to chondrocyte protection. However, to verify this hypothesis, further study on the roles of ANG II in chondrocytes is required.

## 4. Materials and Methods

### 4.1. Materials

Cell culture dishes and multi-well plates were purchased from Becton Dickinson Biosciences (Bedford, MA, USA), Thermo Scientific (Waltham, MA, USA), and TrueLine (Nippon Genetics Co. Ltd.; Tokyo, Japan). Dulbecco’s modified Eagle’s medium (DMEM) and alpha modified Eagle’s medium (αMEM) were purchased from Nissui Pharmaceutical Co., Ltd. (Tokyo, Japan) and ICN Biomedicals (Aurora, OH, USA), respectively, and fetal bovine serum (FBS) was from Nichirei Bioscience Inc. (Tokyo, Japan). Angiotensin II was from Peptide Institute Inc. (Osaka, Japan). Losartan potassium salt and PD123319 were purchased from Fujifilm Wako Pure Chemical (Osaka, Japan) and Abcam (Cambridge, UK), respectively. MIA was obtained from Sigma-Aldrich (St. Louis, MO, USA).

### 4.2. Isolation of Chondrocytes from the Xiphoid Process

To isolate chondrocytes, we isolated the xiphoid process from 7-month-old and 13-month-old female mice. Then, the perichondrium was mechanically removed by using a scalpel, and the xiphoid process was treated with 0.2% collagenase (Fujifilm Wako Pure Chemical) [29]. The liberated chondrocytes were seeded at a density of 1 × 10^4^ cells/cm^2^ and cultured in αMEM containing 10% FBS at 37 °C in a humidified atmosphere of 5% CO_2_. The Animal Committee of Okayama University Graduate School of Medicine, Dentistry, and Pharmaceutical Sciences approved the procedures.

### 4.3. Cell Cultures

A human chondrosarcoma-derived chondrocyte cell line, HCS-2/8 [27], and a rat chondrosarcoma cell line, RCS [28], were inoculated at a density of 4 × 10^4^ cells/cm^2^ and 2 × 10^4^ cells/cm^2^, respectively, into culture dishes containing DMEM supplemented with 10% FBS and cultured at 37 °C in humidified atmosphere of 5% CO_2_.

### 4.4. Generation of Agtr1a-Deficient RCS Cells by Using the CRISPER-Cas9 System

To analyze the effect of *Agtr1a* deletion, we generated *Agtr1a*-deficient RCS cells using a genome editing technique with the CRISPER-Cas9 system. Two guide RNAs were synthesized by GeneArt^TM^ Precision gRNA Synthesis Kit (Thermo Fisher Scientific). Primer sequences to synthesize cDNA for the guide RNAs are indicated in Table 1. After RCS cells had reached confluence, the cells were collected with 0.25% trypsin-EDTA, and 1 × 10^6^ cells were re-suspended in Nucleofactor solution containing 50 μg of Cas9 protein (Integrated DNA Technologies, Coralville, IA, USA) and both gRNAs. Electroporation was performed using an Amaxa^TM^ Human Chondrocyte Nucleofector^TM^ kit and Amaxa Nucleofector1 II (Lonza Cologne GmbH, Cologne, Germany) according to the manufacturer’s instructions [31]. The next day, single cell clones were selected through serial dilution, and the deletion of *Agtr1a* in each clone was verified by genomic PCR analysis, followed by DNA sequencing.

### 4.5. Quantitative Reverse Transcription (RT)-PCR Analysis

To isolate total RNA from HCS-2/8 cells, RCS cells, and xiphoid process-derived chondrocytes, we used ISOGEN reagent (Nippon Gene, Tokyo, Japan). First-strand cDNA was synthesized from 1 μg of total RNA by a primerScript^TM^ RT reagent kit (Takara Shuzo, Tokyo, Japan), and subsequent quantitative PCR analysis was performed with the SYBR^®^ Green Real-time PCR Master Mix (Toyobo, Tokyo, Japan) using the StepOne Plus real-time PCR machine (Applied Biosystems, Carlsbad, CA) as described previously [32]. The specific primer sequences and the GenBank accession numbers of the target genes are shown in Table 2.

### 4.6. Western Blot Analysis

After the cell lysate was extracted with cell lysis buffer (20 mM Tris-HCl pH 7.7, 150 mM NaCl, 1 mM EDTA, 1% Triton X-100) from HCS-2/8 and RCS cells, the concentration of proteins in the cell lysates were determined by the BCA protein assay kit (Thermo Fisher Scientific) using a bovine serum albumin (BSA) standard curve. Then, Western blot analysis was performed as described previously [31]. The following primary antibodies were used: mouse anti-AT_1_ receptor antibody (Santa Cruz Biotechnology, Santa Cruz, CA, USA), rabbit anti-CCN2 antibody (Abcam), rabbit anti-MMP9 antibody (Triple Point Biologics, Inc., Forest Grove, OR, USA), rabbit anti-ERK1/2 antibody (Cell Signaling Technology, Beverly, MA, USA), rabbit anti-phospho-ERK1/2 antibody (Promega, Madison, WI, USA), and mouse anti-β-actin antibody (Sigma-Aldrich). The following secondary antibodies were used: horseradish peroxidase (HRP)-conjugated anti-rabbit IgG antibody (SeraCare Life Sciences, Inc., Milford, MA, USA) or HRP-conjugated anti-mouse IgG antibody (Rockland, Limerick, PA, USA).

### 4.7. Cell Proliferation Assay

For the evaluation of cell proliferation, we used Cell Counting Kit-8 (CCK-8) according to the recommended protocol (DOJINDO, Kumamoto, Japan). Briefly, 10 μL of CCK-8 solution was added to 100 μL of culture media of RCS cells treated with ANG II. After incubation for 45 min at 37 °C, the absorbance of the culture medium was measured at a wavelength of 450 nm [33].

### 4.8. Measurement of Sulfated Glycosaminoglycan (GAG) Content

RCS cells were treated with ANG II at concentrations of 100 and 250 nM for 7 days immediately after being inoculated into a 24-well plate. Thereafter, the cell lysate was collected and digested with actinase E for 90 min at 65 °C. DMB (1, 9-dimethyl methylene blue) solution was added to actinase E-digested cell lysate, and optical absorbance was measured at 540 nm by using diluted chondroitin-4-sulfate as the standard solutions [33].

### 4.9. Gelatin Zymography

For the measurement of the gelatinase activity of ANG II-induced MMP9, we performed gelatin-substrate zymography using the culture medium from RCS cells treated with ANG II, as described previously [34]. Briefly, 500 μL of cultured medium was concentrated 10 times with the Amicon® Ultra-0.5 centrifugal filter devices (EMD Millipore, Temecula, CA, USA). Then, 10 μL of this solution was electrophoresed at 4 °C in a 7.5% polyacrylamide gel containing 0.1% gelatin. After electrophoresis, the gel was washed twice with 2.5% Triton X-100 and incubated at 37 °C for 24 h in 50 mM Tris-HCl pH 8.0, 0.5 M CaCl_2_, and 10^−6^ M ZnCl_2_. The gel was stained with 1% Coomassie brilliant blue.

### 4.10. Immunohistochemistry

Mice at 7 and 13 months of age were euthanized to obtain knee joints. The knee joints were fixed in 10% formalin overnight at 4 °C and decalcified with 10% formic acid before being embedded in paraffin [35]. Five-micrometer sections were mounted on glass slides and deparaffinized with xylene, rehydrated, and treated with hyaluronidase (25 mg/mL; Sigma) for 30 min at room temperature for antigen retrieval. Immunohistochemical staining of these sections was performed with rabbit anti-AT_1_ receptor (AT_1_R) antibody (EMD Millipore), as described previously [35]. Sections stained with a normal rabbit antibody (Dako, Tokyo, Japan) were used as negative controls.

### 4.11. Statistical Analysis

All experiments were repeated at least twice, and similar results were obtained. After performing the F-test (normality testing), statistical analyses for multiple comparisons were performed by one-way analysis of variance (ANOVA) and Bonferroni’s test. Statistical analyses for two groups were performed by unpaired Student’s *t*-test. All data are shown as the mean and standard deviation (SD).

## Figures and Tables

**Figure 1 ijms-22-09204-f001:**
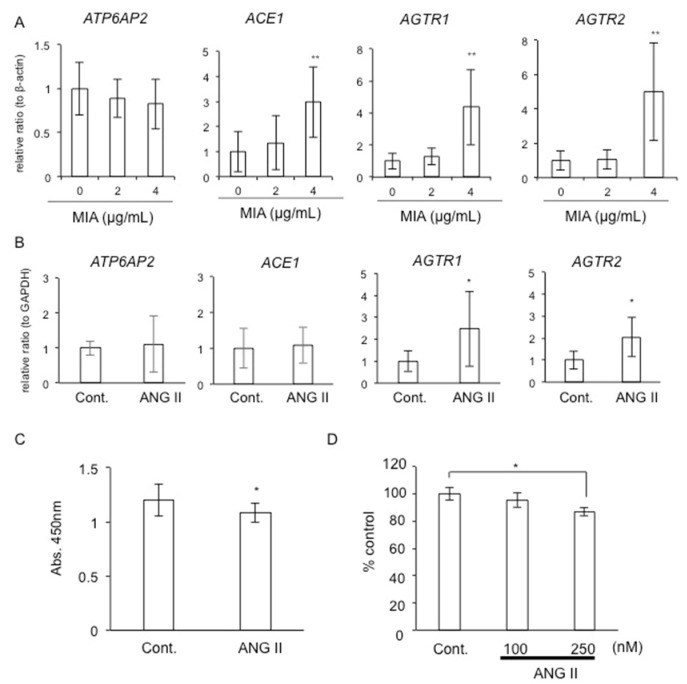
Decreased chondrocyte proliferation and differentiation by ANG II. (**A**) The gene expression of RAS components in HCS-2/8 cells treated with MIA. After HCS-2/8 cells had reached confluence, the cells were treated with MIA at concentrations of 2 and 4 μg/mL for 12 h. Quantitative RT-PCR analysis was performed, and the amounts of these mRNAs were normalized to the amount of *β-actin* mRNA. The ordinates of the graphs indicate the relative ratio with respect to the control (ratio = 1.0). (**B**) Effect of ANG II on the gene expression of RAS components in HCS-2/8 cells. After HCS-2/8 cells had reached confluence, the cells were treated with 200 nM ANG II for 24 h. Graphs show the relative ratio with respect to the control (ratio = 1.0). (**C**) Cell proliferation assay of RCS cells treated with ANG II. RCS cells were inoculated into 96-well plates and stimulated with 200 nM ANG II at the same time. After 48 h, WST-8 was added, and the optical absorbance of culture media was measured at a wavelength of 450 nm. Each bar shows the mean and standard deviation of the results from 10 wells. (**D**) Accumulation of GAGs in RCS cells treated with ANG II. RCS cells were inoculated and stimulated with ANG II at the indicated concentrations. After 4 days, the culture media were changed, and the cells were stimulated with ANG II for another 3 days. Then, the cell lysate was prepared, and GAG content was measured. The ordinates of the graphs indicate the relative ratio with respect to the control (ratio = 100%). The data were analyzed by using Bonferroni’s test to compare the data from multiple groups or unpaired Student’s *t*-test to compare the data from 2 groups. The asterisk indicates a significant difference from the control (* *p* < 0.05, ** *p* < 0.01).

**Figure 2 ijms-22-09204-f002:**
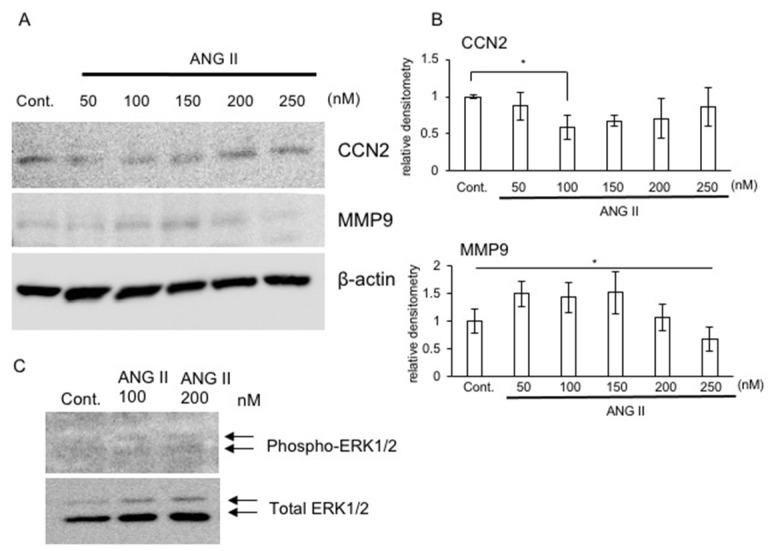
(**A**) Effect of ANG II on the production of CCN2 and MMP9 in HCS-2/8 cells. HCS-2/8 cells were treated with ANG II at the indicated concentrations for 24 h. Then, Western blot analysis was performed. (**B**) The amounts of CCN2 and MMP9 were determined densitometrically and these amounts were normalized to the amount of β-actin. Relative densitometry (control = 1.0) from 4 measurements is presented and was analyzed by Bonferroni’s test and one-way ANOVA. The asterisk indicates a significant difference from the control at * *p* < 0.05. (**C**) Activation of ERK-1/2 in RCS cells treated with ANG II. RCS cells were treated with ANG II at the indicated concentrations for 15 and 30 min. Western blot analysis was performed with antibodies against the indicated proteins.

**Figure 3 ijms-22-09204-f003:**
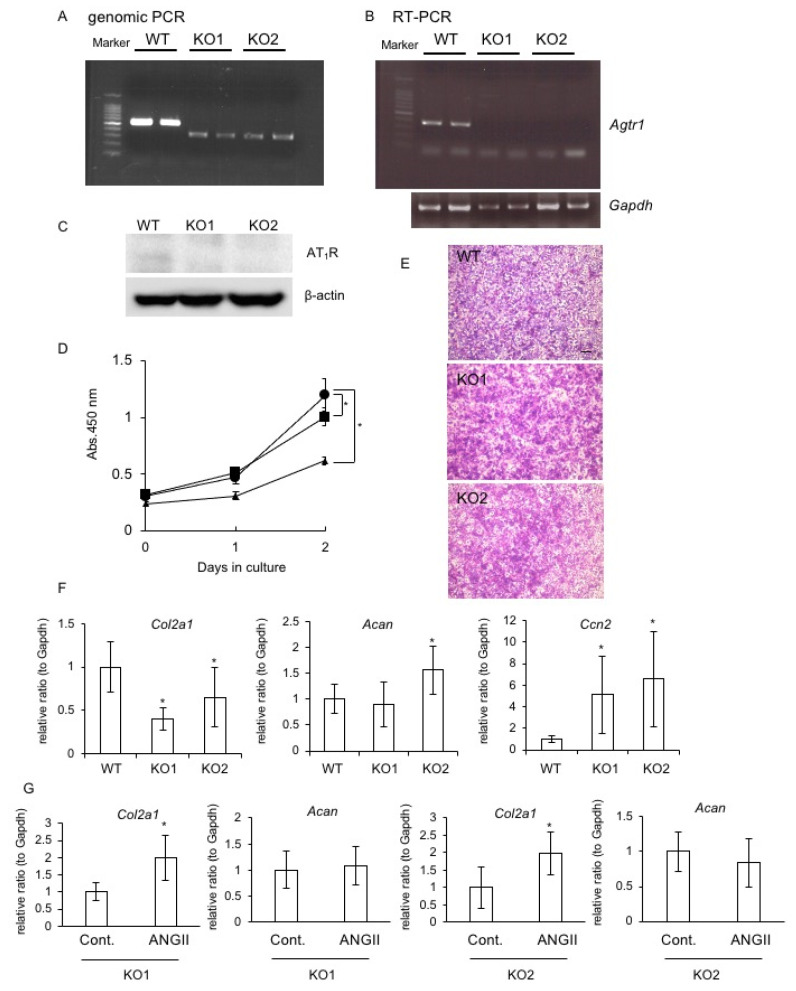
Generation of *Agtr1a*-deficient RCS cells. (**A**) Agarose gel electrophoresis shows PCR products of 526 base pairs (bp) from WT cells and 290 bp from KO1 and KO2 cells. The bands are shown in duplicate. (**B**) *Agtr1a* expression in WT, KO1, and KO2 cells at the mRNA level. The bands are shown in duplicate. (**C**) Western blot analysis of AT_1_R in WT, KO1, and KO2 cells. The band of AT_1_R was detected in WT cells but not in KO1 and KO2 cells. (**D**) Cell proliferation of WT (closed circle), KO1 (closed square), and KO2 cells (closed triangle) determined by using WST-8 on days 0, 1, and 2. The graph shows the mean and standard deviation of the results from 10 wells. (**E**) Toluidine blue staining of WT, KO1, and KO2 cells at confluence. Metachromasia was more pronounced in KO1 and KO2 cells than it was in WT cells. Scale bar = 100 μm. (**F**) The gene expression of *Col2a1*, *Acan,* and *Ccn2* in WT, KO1, and KO2 cells by quantitative RT-PCR analysis. The amounts of these mRNAs were normalized to the amount of *GAPDH* mRNA. (**G**) Effect of ANG II on the gene expression of *Col2a1* and *Acan* in KO1 and KO2 cells. KO1 and KO2 cells were treated with 200 nM ANG II for 24 h. The ordinates of the graphs indicate the relative ratio with respect to the control (control = 1.0). The data were analyzed by using the unpaired Student’s *t*-test to compare the data from 2 groups. The asterisk indicates a significant difference from the WT at * *p* < 0.05.

**Figure 4 ijms-22-09204-f004:**
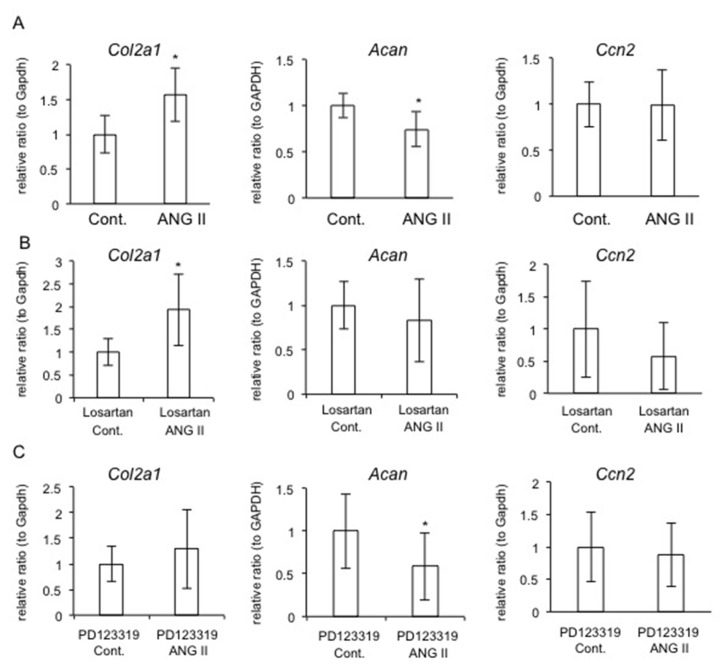
Effect of AT_1_R and AT_2_R blockade on the gene expression of *Col2a1*, *Acan,* and *Ccn2* in RCS cells. (**A**) RCS cells were treated with ANG II at 200 nM for 24 h. (**B**) RCS cells were pre-treated with 10 μM losartan potassium, an AT_1_R antagonist, for 30 min and were stimulated with ANG II at 200 nM for 24 h. (**C**) RCS cells were pre-treated with 10 μM PD123319, an AT_2_R antagonist, for 30 min and were stimulated with ANG II at 200 nM for 24 h. Quantitative RT-PCR analysis was performed. The amounts of these mRNAs were normalized to the amount of *Gapdh* mRNA, and the relative ratios with respect to the control (ratio = 1.0) are indicated. The asterisk indicates a significant difference from the control at * *p* < 0.05.

**Figure 5 ijms-22-09204-f005:**
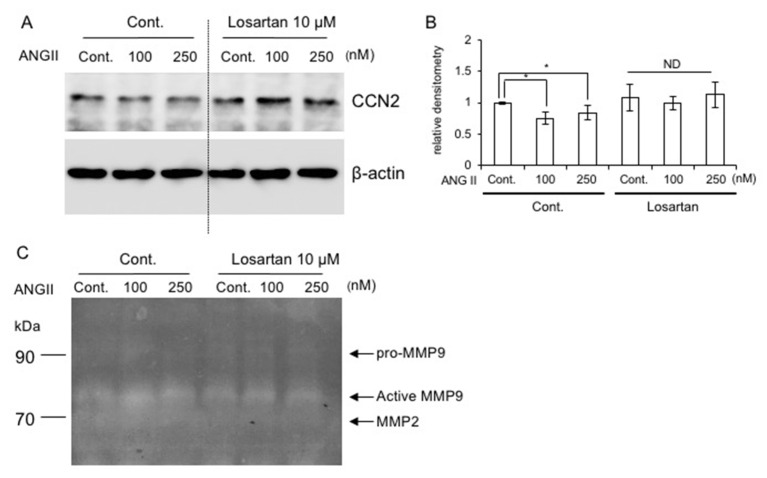
Effect of losartan on the production of CCN2 in RCS cells. (**A**) RCS cells were pre-treated with 10 μM losartan potassium for 30 min and were stimulated with ANG II at 100 and 250 nM for 24 h. (**B**) The amount of CCN2 was determined densitometrically, and these amounts were normalized to the amount of β-actin. Relative densitometry (control without losartan = 1.0) is presented and was analyzed by Bonferroni’s test. The asterisk indicates a significant difference from the control at * *p* < 0.05. ND represents “no difference”. (**C**) Effect of losartan on secreted gelatinases in the culture medium from RCS cells treated with 100 and 250 nM ANG II for 24 h. Gelatin-substrate zymogram was performed.

**Figure 6 ijms-22-09204-f006:**
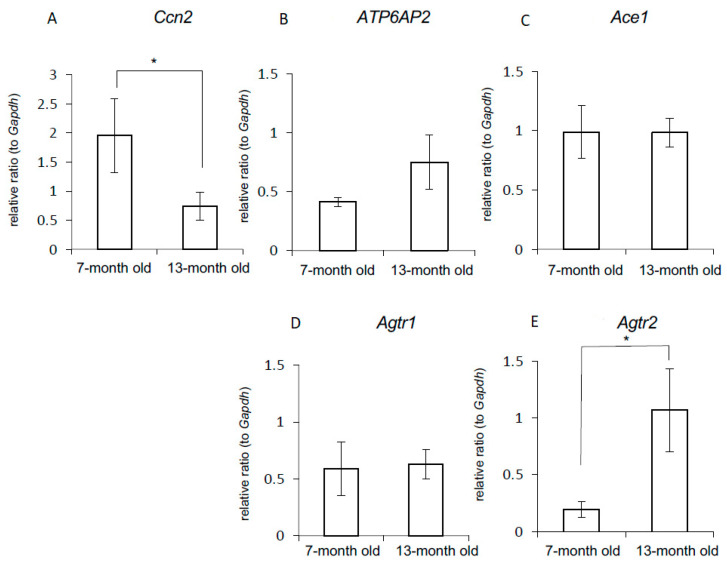
Change in the gene expression of CCN2 and RAS components accompanied by aging. Chondrocytes were isolated from the xiphoid process of mice at 7 and 13 months of age. These cells were grown until they reached confluence. Then, total RNA was isolated, and quantitative RT-PCR analysis was performed by using the indicated primers. The amounts of these mRNAs were normalized to that of *Gapdh* mRNA. The graphs show the expression levels of (**A**) *Ccn2*, (**B**) *Atp6ap2*, (**C**) *Ace1,* (**D**) *Agtr1,* and (**E**) *Agtr2.* The asterisk indicates a significant difference (* *p* < 0.05). The data were analyzed by using the unpaired Student’s *t*-test.

**Figure 7 ijms-22-09204-f007:**
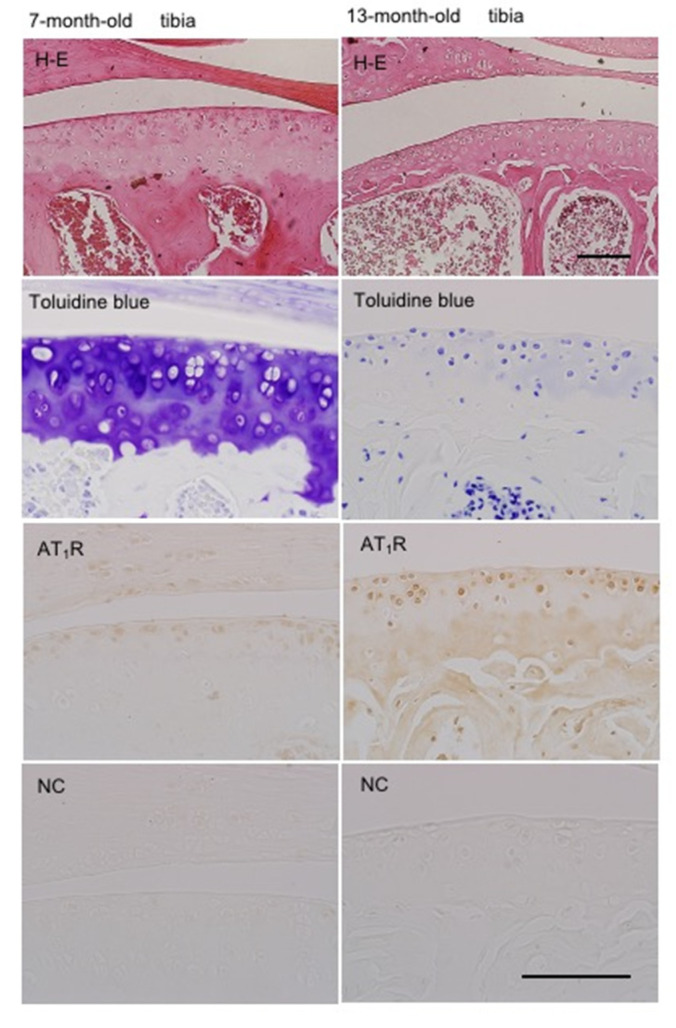
Immunohistochemical analysis of articular cartilage from mice at 7 and 13 months of age, using rabbit anti-AT_1_R antibody. Sections of the articular cartilage of tibias were stained with hematoxylin and eosin (H&E) and toluidine blue. Cartilage tissue of 7-month-old tibias had a metachromatic appearance with toluidine blue staining, and AT_1_R was detected on the surface of articular cartilage. On the other hand, the cartilage tissue of 13-month-old tibias did not have a metachromatic appearance after toluidine blue staining, but AT_1_R was detected throughout the articular cartilage. Negative controls (NC) showed no detectable signals. Scale bars = 100 μm.

**Figure 8 ijms-22-09204-f008:**
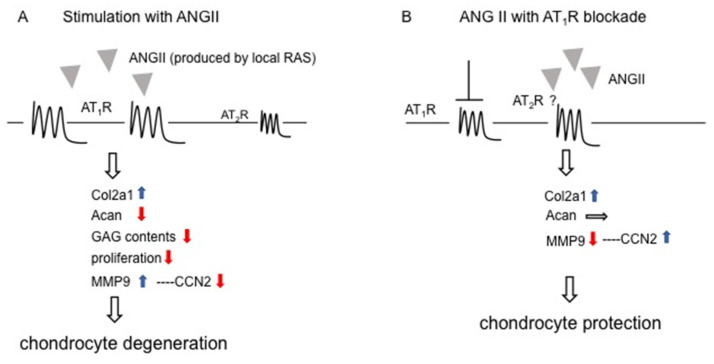
Schematic representation of possible chondrocyte degeneration and protection by the ANG II–AT_1_R axis (**A**) and AT_1_R blockade (**B**), respectively. (**A**) Since AT_1_R is the dominant ANG II receptor in chondrocytes, the ANG II–AT_1_R axis decreases not only *Acan* expression and GAG accumulation but also the CCN2 level via increased MMP9, leading to chondrocyte degeneration. (**B**) When AT_1_R is blocked by a specific inhibitor, such as losartan, ANG II interacts with AT_2_R instead. The ANG II–AT_2_R axis then increases the CCN2 level via decreased MMP9, leading to chondrocyte protection. Gray triangles indicate ANG II. Blue and red arrows show upregulation and downregulation, respectively, and horizontal arrow indicates no effect. The question mark represents the possible interaction of ANG II with AT_2_R, when AT_1_R is blocked by a specific inhibitor.

**Table 1 ijms-22-09204-t001:** Sequences of (F) and (R) primers used for genome editing of rat *Agtr1a*.

Application	Primer Sequences	Expected Length (bp)	
		WT	KO
Guide RNA1	(F) 5′-TAATACGACTCACTATAGCGATGAAGTATACATTTCGG-3′ (R) 5′-TTCTAGCTCTAAAACCCACCGAAATGTATACTTCATCG-3′		
Guide RNA2	(F) 5′-TAATACGACTCACTATAGTGGGCAGTCTATACCGCTA-3′ (R) 5′-TTCTAGCTCTAAAACTAGCGGTATAGACTGCCCA-3′		
Genomic PCR	(F) 5′-CAAGCGTCTTTCTTCTCAATCTC-3′ (R) 5′-GATGTCATCGTTTCTTGGTTTG-3′	526	290
RT-PCR	(F) 5′-TGGGCAGTCTATACCGCTA-3′ (R) 5′-GCCCTATGGGGAGCGTCGAA-3′	334	ND

ND: not detected.

**Table 2 ijms-22-09204-t002:** Sequences of forward (F) and reverse (R) primers used for quantitative PCR.

Gene	Accession No.	Species	Primer Sequences	PCR Product Length (bp)
*GAPDH*	XM_011241214.1	human /mouse	(F) 5′-GCCAAAAGGGTCATCATCTC-3′ (R) 5′-GTCTTCTGGGTGGCAGTGAT-3′	214
*ATP6AP2*	NM_005765.3	human	(F) 5′-CCTCATTAGGAAGACAAGGACTATCC-3′ (R) 5′-GGGTTCTTCGCTTGTTTTGC-3′	50
*ACE1*	NM_000789.4	human	(F) 5′-AAGCAGGACGGCTTCACAGA-3′ (R) 5′-GGGTCCCCTGAGGTTGATGTAT-3′	182
*AGTR1*	NM_009585.4	human	(F) 5′-CTTCAGCCAGCGTCAGTT-3′ (R) 5′-TGCAGGTGACTTTGGCTA-3′	137
*AGTR2*	NM_000686.5	human	(F) 5′-ACTTCGGGCTTGTGAACATC-3′ (R) 5′-TAAATCAGCCACAGCGAGGT-3′	217
*ACTB*	NM_001101.3	human	(F) 5′-GATCATTGCTCCTCCTGAGC-3′ (F) 5′-ACTCCTGCTTGCTGATCCAC-3′	100
*Acan*	XM_039101035.1	rat	(F) 5′-GATGTCCCCTGCAATTACCA-3′ (R) 5′-TCTGTGCAAGTGATTCGAGG-3′	229
*Col2a1*	NM_012929.1	rat	(F) 5′-CCCAGAACATCACCTACCAC-3′ (R) 5′-GGTACTCGATGATGGTCTTG-3′	200
*Ccn2*	NM_022266.2	rat	(F) 5′-ATCCCTGCGACCCACACAAG-3′ (R) 5′-CAACTGCTTTGGAAGGACTCGC-3′	144
*Gapdh*	NM_017008.4	rat	(F) 5′-GCCAAAAGGGTCATCATCTC-3′ (R) 5′-GTCTTCTGAGTGGCAGTGAT-3′	214
*CCN2*	NM_010217.2	mouse	(F) 5′-CCACCCGAGTTACCAATGAC -3′ (R) 5′-GTGCAGCCAGAAAGCTCA-3′	151
*Atp6ap2*	NM_027439.4	mouse	(F) 5′-CACATTGCGGCAGCTCCGTAA-3′ (R) 5′-CTCACAAGGGATGTGTCGAAT-3′	377
*Ace1*	NM_207624.6	mouse	(F) 5′-AACAAACATGATGGCCACATCCCG-3′ (R) 5′-CGTGTAGCCATTGAGCTTGGCAAT-3′	147
*Agtr1*	NM_177322.3	mouse	(F) 5′-CCATTGTCCACCCGATGAAG-3′ (R) 5′-TGCAGGTGACTTTGGCCAC-3′	62
*Agtr2*	NM_007429.5	mouse	(F) 5′-CAGCAGCCGTCCTTTTGATAA-3′ (R) 5′-TTATCTGATGGTTTGTGTGAGCAA -3′	80

## Data Availability

We did not report any supplementary data.

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
