# Peer review of "Effect of Angiotensin II on Chondrocyte Degeneration and Protection via Differential Usage of Angiotensin II Receptors"

_ijms, 2021, doi:10.3390/ijms22179204_

Round 1

Reviewer 1 Report

The manuscript by Nishida et al. demonstrated that treatment with angiotensin II (Ang II) regulated cartilage degeneration. The results are interesting, and the manuscript was well written. Some minor comments are existed in the present study.

The authors speculated AT1R blockade reduces chondrocyte degeneration induced by ANG II via the other receptor such as AT2R. How about the effect of PD123319, an AT2R antagonist, on the losartan-treated RCS cells? It is better to prove the mechanism described in Figure 7b.

Are there some clinical indications that blockade of the renin-angiotensin system affects cartilage-related diseases? Please add them in the discussion.

Author Response

Responses to Reviewer#1

We thank the Reviewer for precious and helpful suggestions.  Addressing the reviewer’s comments, we have revised the manuscript as indicated by reviewer, and our responses regarding the specific points are summarized below. 

Reviewer comment:

The text style and grammar of the manuscript need substantial improvement.

Author response and action

We apologize for the grammatical errors in our original manuscript.  We have checked our manuscript carefully and we have re-written our manuscript.  Furthermore, we had our final manuscript checked by a professional English editor.  A certificate for English editing is attached.

Reviewer comment:

They need to show the effect of an AT2R receptor antagonist.

Author response and action

According to the reviewer's suggestion, we have newly performed additional experiments.  We obtained PD123319, an AT2R antagonist and investigated the effect of angiotensin II on Col2a1 and Aggrecan expressions in RCS cells pre-treated with PD123319.  As a result, the gene expression of Col2a1 was increased by treatment with Losartan, an AT1R antagonist, and Aggrecan expression was decreased in the treatment with PD123319.  Ccn2 expression remained unchanged.  There results have been added as Figure 4 in the revised version. 

Reviewer comment:

They need to show the effect of an AT1R blocker in the absence or presence of and AT2R antagonist in aged and/or obese rats. 

Author response and action

The reviewer has pointed out very important points in this study.  Therefore, we investigated the effect of Losartan in the absence or presence of and PD123319 on chondrocyte differentiation.  These results have been added as Figure 4C in the revised version.  However, preparation of aged or obese rats and isolation of chondrocytes from these animals need much longer time than the period allowed for the revision.  Thus, this important point will be addressed in a future study.

Reviewer 2 Report

Nishida et al. describe in this manuscript (ijms-1292864) the biphasic effects of Ang II on chondrocyte degeneration and protection via diverse Ang II receptors.   we investigated how ANG II and AT1R blockade affect chondrocyte proliferation and differentiation. Firstly, we showed that ANG II significantly suppressed cell proliferation and glycosaminoglycan content in rat chondrocytic RCS cells. Additionally, ANG II decreased cellular communication network factor 2 (CCN2), which is an anabolic factor for chondrocytes, via increased MMP9. In Agtr1a-deficient RCS cells generated by CRISPR-Cas9 system, Ccn2 and Aggrecan expressions were increased. In RCS cells treated with Losartan, an AT1 receptor antagonist, CCN2 was not decreased by Ang II treatment.  They concluded that Ang II regulates age-related cartilage degeneration through AT1 receptors.  The text style and grammar of the manuscript need substantial improvement.  They need more robust data before yielding a solid conclusion.  They need to show 1) the effect of an AT2 receptor antagonist, and 2) the effect of an AT1 blocker in the absence or presence of and AT2 antagonist in aged and/or obese rats to provide more strength and significance to the study.

Comments:

  1. Please spell out RCS and HCS-2/8, and what’s the difference between the 2 cell-types? And what’s the rationale of using different cells for different experiments?

  1. Is expression of AT1 and AT2 receptors affected by obesity?  What is the status of renin -angiotensin system in obesity?  Does chronic treatment with an AT1 receptor alleviate the severity of osteoarthritis in obese rodents?  Or, at least, use of chondrocytes from lean versus obese rodents could be of interest. 

  1. They need to show the effect of an AT2 receptor antagonist in the absence or presence of AT1 antagonist in cells treated with vehicle or Ang II.

  1. Inclusion of “prorenin” in the introduction is drifting the attention of the reader since all the work is about the effects of Ang II via AT1 or AT2 receptor on chondrocytes. And, in Figure 2 the author showed biphasic effects of Ang II on CCN2 and MMP9 expression; is this due to the activation of the counter acting AT2 receptors or AT receptor desensitization?  They should show a zymogram gel analysis for MMP2 and MMP9.

  1. Figure 2C, the gel is of poor quality. The 42 and 44 bands for phosphor-ERK are not clear.

  1. Legend for figures need to be re-written by avoiding redundant information. .

  1. There is a need of substantial text style and grammatical improvement of the entire manuscript. For example, 1) lines 20-21; “In RCS cells treated with Losartan, an AT1R inhibitor, CCN2 was not decreased by ANG II treatment” should read as follows: “Losartan, an AT1 receptor antagonist, blocked Ang II-induced CCN2 expression in cultured RCS cells”; 2) lines 97-98; “Next, we examined whether or not the ANG II effected on the gene expression of local RAS components” should read “Next, we examined whether Ang II affected gene expression of local RAS components”; 3) line 108, “On the other hand, Ccn2 expression had no effect (Fig. 1C).” should be “On the other hand, Ccn2 expression remained unchanged when RCS were stimulated with Ang II”…etc. 

  1. How do they explain data from Figure 3. Both KO1 and KO2 delivered total absence of AT1 receptor expression and yet they showed that only KO1 showed absent pharmacologic effects of Ang II?

  1. Data from Figure 9. They should have shown effect of losartan in the absence or presence of AT 2 receptor blocker in aged animals. 

Author Response

Responses to Reviewer#2

Thank you very much for precious and helpful instructions.  Following the reviewer’s comments, we have modified our manuscript accordingly, employed additional experiments and added new data in our revised manuscript.

Reviewer comment:

Please spell out RCS and HCS-2/8, and what’s the difference between the 2 cell-types?  And what’s the rationale of using different cells for different experiments?

Author response and action

Following the reviewer’s comment, we have shown the full names of RCS cells and HCS-2/8 cells (Line 88-89 and line 105 in the revise version), which are detailed in the references (27 and 28). 

HCS-2/8 cell was established from a human chondrosarcoma and is an immortalized chondrocyte that retains the normal human chondrocyte phenotypes, such as COL2a1 and AGGRECAN, during serial passage.  These cells are very precious and useful as an in vitro model of human chondrocytes, but cell growth is slow and thus cell cloning is quite difficult.  RCS cell was also established from rat chondrosarcoma and displays a stable differentiated chondrocyte-like phenotype.  Importantly, cell growth of RCS cells is faster than that of HCS-2/8 cells, which enables efficiently cell cloning.  We initially employed HCS-2/8 cells as a human model.  However, in order to generate Agtr1a-deficient cell clones, we needed to use RCS cells, and used RCS cells in relevant experiments.    

Reviewer comment:

Is expression of AT1 and AT2 receptors affected by obesity?  What is the status of renin-angiotensin system in obesity?  

Author response and action

Obesity is a complex disease, and it is a risk factor for various chronic diseases including breast cancer, OA, and hypertension.  It is well-known that adipose tissue produces angiotensin II, and that the expression of its receptor is affected by obesity, suggesting the involvement of renin-angiotensin system in obesity.  In fact, a review article describing the mechanisms linking the renin-angiotensin system, obesity, and breast cancer was recently published (ref 3).  We have added this review to the references for the interest of readers.

Reviewer comment:

Does chronic treatment with an AT1 receptor alleviate the severity of osteoarthritis in obese rodents?

Author response and action

The reviewer has pointed out very important point in this study.  Based on this study, we believe that AT1R antagonist alleviate the severity of OA.  We are going to verify this hypothesis in our next study. 

Reviewer comment:

They need to show the effect of an AT2 receptor antagonist in the absence or presence of AT1 antagonist in cells treated with vehicle or ANG II.

Author response and action

According to the reviewer’s suggestion, we have newly performed additional experiments using PD123319, an AT2R antagonist.  We have shown these results as new Figure 4 in the revised version. 

Reviewer comment:

Inclusion of “prorenin” in the introduction is drifting the attention of the reader.

Author response and action

According to the reviewer's suggestion, we have rewritten “Introduction” section, deleting the background of discovery of prorenin. 

Reviewer comment:

The author showed biphasic effects of ANG II on CCN2 and MMP9 expression; is this due to the activation of the counter acting AT2 receptors or AT1 receptor desensitization?

Author response and action

As the review pointed out, AT1 receptor desensitization could result in such biphasic effect on CCN2 and MMP9 in the presence of pre-existing ANG II.  If so, Col2a1 should have been repressed and Acan should have been induced at the same time.  However, Col2a1 was induced and Acan was unchanged upon AT1 blockade, suggesting the activation of a distinct signaling by the other receptor.  Furthermore, Ccn2 expression remained unchanged when RCS cells were treated with AT1R antagonist alone, indicating the absence of pre-existing ANG II.  Taken together with these results, CCN2 production is potentially modulated via MMP9 increased or decreased by ANG II.  Therefore, we consider biphasic effects of ANG II is due to the activation of the counter acting AT2 receptors.

Reviewer comment:

They should show a zymogram gel analysis for MMP2 and MMP9.

Author response and action

Following the reviewer’s comment, we have performed gelatin zymography using conditioned medium from RCS cells treated with ANG II.  This result is shown as Figure 5C in the revised version.  

Reviewer comment:

Figure 2C, the gel is of poor quality. The 42 and 44 bands for phospho-ERK are not clear.

Author response and action

Agreeing the reviewer’s comment, we have re-performed Western blot analysis to detect the 42 and 44 bands for phospho-ERK and total ERK.  These results have shown as Figure 2D in the revised version.

Reviewer comment:

Legend for figures need to be re-written by avoiding redundant information.

Author response and action

Following the reviewer’s suggestion, we have deleted redundant information and have rewritten all Figure legends. 

Reviewer comment:

There is a need of substantial text style and grammatical improvement of the entire manuscript. 

Author response and action

Thank you for the advice.  We have checked, and re-written the entire manuscript carefully.  Furthermore, we had our final version checked by a professional English editor.  A certificate for English editing is attached.   

Reviewer comment:

How do they explain data from Figure 3.  Both KO1 and KO2 delivered total absence of AT1receptor expression and yet they showed that only KO1 showed absent pharmacologic effects of ANG II?

Author response and action

As shown in Figure 3, Agtr1a expression was not detected either in KO1 or KO2 cells. Cell proliferation was suppressed, whereas the metachromatic properties of toluidine blue staining were enhanced in both KO1 and KO2 cells.  Finally, KO1 and KO2 cells showed comparable effects of ANG II on the gene expression of Col2a1 and Acan. These results indicate that both KO1 and KO2 cells show quite similar pharmacologic effects of ANG II.  However, the gene expression of Acan in KO2 cells was up-regulated, while it was not in KO1 cells (Figure 3F).  It is not clear what made this difference.  We suspect that minor variation in the nucleotide sequence formed through the CRISPR-Cas9-mediated genome editing may have had a certain impact on the transcriptomic landscape. 

Reviewer comment:

They should have shown effect of Losartan in the absence or presence of AT2 receptor blocker in aged animals. 

Author response and action

The reviewer has pointed out very important issue in this study.  Therefore, we investigated the effect of AT2R antagonist on chondrocyte differentiation.  These results have added as new Figure 4C in the revised version.  However, the period given for the revision was too short to obtain chondrocytes from aged animals.  This experiment will be saved for our future study. 
